# An osteoporosis course as a separate component of problem-based learning

Iva Hoffmanová[1]*, Valér Džupa[2], Petr Waldauf[3], Robert Grill[4‡], Václav Báča[5‡]

**1** Department of Internal Medicine, Second Faculty of Medicine, Charles University, Prague, Czech Republic, **2** Department of Orthopedics and Traumatology, Third Faculty of Medicine, Charles University, Prague, Czech Republic, **3** Department of Anesthesiology and Resuscitation, Third Faculty of Medicine, Charles University, Prague, Czech Republic, **4** Department of Urology, Third Faculty of Medicine, Charles University, Prague, Czech Republic, **5** Department of Anatomy, Third Faculty of Medicine, Charles University, Prague, Czech Republic

☺ These authors contributed equally to this work.
‡ RG and VB also contributed equally to this work.
* iva.hoffmanova@gmail.com

## Abstract

The objective was to compare the extent of acquired knowledge regarding osteoporosis-related issues in a group of medical students who successfully completed an optional "Elective Osteoporosis Course" based on problem-based learning, and a group of medical students who completed only the mandatory curriculum.

## Study groups and methods

Study group I was comprised of 25 fourth-year students who successfully completed the Elective Osteoporosis Course I (focused on pathophysiology, diagnostics, and pharmacological treatment), while control group I was comprised of 25 fifth-year students who successfully completed only all required fourth- and fifth-year courses, but did not participate in the elective Osteoporosis I course. Study group II was comprised of 27 fourth-year students who successfully completed the Elective Osteoporosis Course II (focused on treatment of osteoporotic fractures), while control group II was comprised of 24 sixth-year students who were preparing for final exams in surgical disciplines, but did not participate in the Elective Osteoporosis Course II. The groups were compared using a linear regression model with robust estimation of standard errors using Stata 13.1. A p-value < 0.05 was considered statistically significant.

## Results

Study Group I scored, on average, 6.7 points higher than Control Group I on the multiple-choice test (scale –16 to +21). Study Group II scored, on average, 3.5 points higher than Control Group II on the multiple-choice test (scale –21 to +28).

**Data availability statement:** All relevant data are within the paper and its Supporting Information files.

**Funding:** The author(s) received no specific funding for this work.

**Competing interests:** NO authors have competing interests.

Both differences were statistically significant ($p < 0.001$, $R^2 = 0.64$, 95% CI: 5.2–8.1; $p = 0.006$, $R^2 = 0.15$, 95% CI: 1.1–5.9; respectively).

## Conclusion

Results demonstrated a greater understanding in students who participated in problem-based learning medical studies relative to those who completed only the mandatory curriculum.

## Introduction

Osteoporosis represents a growing global healthcare challenge, affecting approximately 21.2% of women and 6.3% of men over the age of 50 worldwide. Osteoporotic fractures, especially hip fractures, are associated with serious complications and a mortality rate of 20–24% within the first year post-fracture. With the global population aging, the incidence of hip fractures is projected to double by 2050, thereby intensifying the burden on healthcare systems worldwide [1].

Despite its clinical significance, osteoporosis remains underrepresented in medical education and clinical practice. Moreover, as a multidisciplinary condition, its teaching is often spread across several study subjects, which can result in students developing a fragmented understanding of the disease. This gap contributes to delayed diagnosis, suboptimal management, and disconnection between theoretical knowledge and its application. Given the increasing prevalence of osteoporosis and osteoporotic fractures worldwide, a more comprehensive integration of this condition into medical curricula is warranted.

At our Faculty of Medicine, we introduced an optional course called the Elective Osteoporosis Course, which is based on the principles of problem-based learning (PBL). This structure enables students to choose topics of interest while ensuring coverage of essential material.

PBL in pre-clinical medical studies is considered an effective method for integrating theoretical and clinical knowledge [2–4]. Elective courses on interdisciplinary topics, such as osteoporosis, can, through PBL, integrate and consolidate educational content that is otherwise dispersed across multiple mandatory courses [5–12].

At our faculty, osteoporosis-related topics are spread across five mandatory courses within the compulsory curriculum: (1) Diagnostics based on imaging techniques; (2) Hematology and Oncology; (3) Diabetology, Endocrinology, Gastroenterology, and Abdominal Surgery; (4) Nephrology, Urology, Rheumatology, and Geriatrics; and (5) Orthopedics, Traumatology, Anesthesiology and Resuscitation. The Elective Osteoporosis Courses were designed to consolidate existing content from the aforementioned mandatory courses without introducing new material, focusing specifically on the pathophysiology, diagnosis, and management of osteoporosis and osteoporotic fractures. The course was required to be offered as an elective, as the current curriculum did not permit it to be designated as mandatory.

This study aims to assess, based on multiple-choice test scores, whether participation in the PBL-based Elective Osteoporosis Course enhances students'



knowledge of osteoporosis-related issues compared with students who complete only the compulsory curriculum, which is largely grounded in traditional didactic teaching methods.

## Methods

### Course structure

The Elective Osteoporosis Course consisted of two sequential sub-courses:

- Elective Osteoporosis **Course I** focused on definition, clinical presentation, biomechanics, bone metabolism, diagnostic approaches, and pharmacological treatment of osteoporosis.

- Elective Osteoporosis **Course II** addressed disease progression and treatment of typical osteoporotic fractures.

Each sub-course included 14 study hours, combining 10 hours of self-directed e-learning via an online platform (http://osteokurz.lf3.cuni.cz/) and two interactive 2-hour seminars. The e-learning component provided a structured knowledge framework, effectively preparing students for the seminars. The interactive seminars, conducted in PBL format, focused on the discussion and analysis of illustrative clinical cases.

All students participated in the interactive seminars and were divided into five groups beforehand. After completing the e-learning module, each group received a specific clinical case topic, which enabled focused preparation. In each seminar, five cases were discussed, resulting in a total of ten clinical cases per sub-course.

Each case began with the presentation of anamnesis and physical examination findings, followed by the assigned group's proposed diagnostic approach. Their reasoning was subsequently discussed with the other groups and the faculty, who guided the debate and provided feedback aligned with established clinical practice.

In the Elective Osteoporosis Course I, the subsequent steps included the evaluation of diagnostic methods, with attention to differential diagnosis, and the development of management strategies encompassing pharmacological treatment, potential side effects of the treatment, and lifestyle modification. This course, therefore, emphasized the principles of diagnosis and long-term disease management.

In the Elective Osteoporosis Course II, the subsequent steps focused on the evaluation of diagnostic methods for osteoporotic fractures, and the development of management strategies centered on fracture treatment and rehabilitation. This course was thus oriented toward the practical aspects of acute fracture care and subsequent rehabilitation.

At each stage, students articulated the rationale for their decisions, engaged in peer-to-peer discussion, and received faculty evaluation

Building on the solid theoretical foundation established in the preceding e-learning module, the seminar structure allowed students to consolidate their knowledge through practical application, to engage actively in clinical reasoning and collaborative problem-solving, and to benefit from peer-to-pear learning while integrating multiple aspects of patient management. Collectively, these activities contributed to a more comprehensive understanding of osteoporotic care.

### Assessment

Each sub-course was completed by taking a final test comprising 10 multiple-choice questions randomly selected from a pool of 100 items. Unlike a conventional multiple-choice test with simple right/wrong scoring, this assessment used a scaled scoring system. Each answer option was assigned a numerical value rating from strongly positive (highly correct) to strongly negative (incorrect). Highly correct answers yielded strongly positive scores, less accurate but still plausible answers provided smaller positive values, while incorrect choices were penalized with negative scores, ranging from mildly negative for partially incorrect options to strongly negative for clearly erroneous ones. This approach discouraged random guessing and allowed the test to capture how well students distinguished between more and less plausible alternatives. Each question in each set of ten multiple-choice questions was designed to be equivalent in content and to maintain a numerically balanced distribution of response options.

A minimum score of 70% was required to pass the test and, where applicable, to receive course credit. The test was integrated into the e-learning platform.

- For the Elective Osteoporosis Course I test, achievable scores ranged from –16 to +21 points, with the 70% passing threshold set at 10 points. This corresponded to approximately 5 points more than could be obtained by indiscriminately marking all options for all questions.

- For the Elective Osteoporosis Course II test, scores ranged from –21 to +28 points, with the 70% passing threshold set at 14 points. This corresponded to approximately 7 points more than could be obtained by indiscriminately marking all options for all questions. The elevated passing threshold was intentionally established at a slightly greater margin than in Course I to accommodate the broader distribution of potential scores.

In both courses, the passing threshold was determined to ensure that students' performance reflected genuine understanding rather than random chance or exhaustive completion. This type of test is developed, implemented, and validated by experts in medical testing at the Third Faculty of Medicine, Charles University, Prague, Czech Republic.

Both the Study Groups and the Control Groups completed the test under standardized conditions, in a lecture hall using computers via an online platform, under the supervision of the teacher. The test was administered with a time limit of 30 minutes.

## Participants and groups

- **Study Group I**: 25 fourth-year students who voluntarily chose and completed the Elective Osteoporosis Course I.

- **Control Group I**: 25 fifth-year students who completed the mandatory osteoporosis-related coursework in their fourth and fifth year, but did not take the elective course. This group was selected to ensure that the content and scope of the presented and assessed material related to osteoporosis in the mandatory curriculum corresponded to that presented in the Elective Osteoporosis Course I.

- **Study Group II**: 27 fourth-year students who voluntarily chose and completed the Elective Osteoporosis Course II. Of these, 25 students had previously completed Elective Osteoporosis Course I, while 2 students were newly included.

- **Control Group II:** 24 sixth-year students who completed all mandatory courses covering osteoporosis, including fifth-year courses in Orthopedics and Traumatology of the musculoskeletal system, but did not participate in the elective course. At the time of testing, these students were preparing for final examinations in surgical disciplines. This group was selected to ensure that the content and scope of presented and assessed osteoporosis material in the mandatory curriculum corresponded to that covered in the Elective Osteoporosis Course II.

All groups shared exposure to the standard curriculum (based on traditional didactic teaching methods); however, elective participants (Study Groups I and II) received consolidated and focused instruction using PBL.

The elective course's voluntary nature introduces potential self-selection bias, as students opting may have greater baseline interest or motivation. The elective format was chosen because mandatory integration into the existing curriculum was not feasible, and this is recognized as a limitation.

## Ethics statement

The study adhered to institutional guidelines for educational research, and all participants provided verbal informed consent prior to testing. The consent was witnessed by the teacher who supervised the test-taking process, and documented by the fact that all participants answered all test questions



## Statistical analysis

Group comparisons were performed using a linear regression model with robust standard errors (Stata 13.1). A p-value < 0.05 was considered statistically significant. Results are presented as regression coefficients (difference between group means) with 95% confidence intervals and coefficient of determination ($R^2$). Moreover, the Bayes' theorem was utilized to enrich the statistical point of view.

## Results

### Elective osteoporosis course I test results

In Study Group I (n = 25), 24 students (96%) successfully passed the course test and were awarded credit; one student failed to achieve the required minimum score and therefore did not receive credit. In contrast, only one student (4%) in Control Group I (n = 25) passed the test, while the remaining 24 were unsuccessful. Table 1 summarizes the distribution of points earned. As illustrated in S1 Graph, Study Group I scored on average 6.7 points higher than Control Group I, a difference was significant (p < 0.001, $R^2$ = 0.64, 95% CI: 5.2–8.1).

**Table 1. Elective Osteoporosis Course I test: Comparison of points acquired by Study Group I and Control Group I.**

| Number of points acquired | Number of students Study Group I ($N^I$) | Number of students Control Group I ($n^I$) |
|---|---|---|
| 2 | | 2 |
| 3 | 1 | 1 |
| 4 | | 3 |
| 5 | | 6 |
| 6 | | 5 |
| 7 | | 6 |
| 8 | | 1 |
| 9 | | |
| 10 | | |
| 11 | 11 | |
| 12 | 2 | |
| 13 | 5 | 1 |
| 14 | 1 | |
| 15 | 1 | |
| 16 | 1 | |
| 17 | 3 | |
| Total number of students | 25 | 25 |

The test data presented above illustrate the impact of PBL using basic statistical analysis. The testing methodology assumes random data selection, with no cross-references between input variables and process conditions.

The effectiveness of PBL can be further evaluated through more advanced statistical methods – for example, by applying Bayes' theorem of conditional probability, where the condition is participation in the course. The positive results of the test under the condition of attending the PBL course represent a conditional probability. This probability, denoted as P(test_positive | attending_PBL), can be calculated as the ratio of the joint probability of attending the PBL course and achieving a positive result, P(attending_PBL ∩ test_positive), to the overall probability of a positive test result, P(test_positive).

P(test_positive | attending_PBL) = P(attending_PBL ∩ test_positive)/ P(test_positive)

Using this approach, the resulting probability of success on the test is 96%. This means that students who completed the Elective Osteoporosis Course I would pass the test with a 96% probability.
Elective Osteoporosis Course II Test Results.

All students in Study Group II (n = 27) successfully passed the test and were awarded credit. In the Control Group II (n = 24), 17 students (71%) passed, while the remaining 7 were unsuccessful. The point distribution is presented in Table 2. Study Group II achieved an average score 3.5 points higher than Control Group II, with this difference also reaching statistical significance (p = 0.006, $R^2$ = 0.15, 95% CI: 1.1–5.9), as shown in S2 Graph

## Discussion

This study evaluated the impact of an Elective Osteoporosis Course combining self-directed e-learning with PBL seminars for fourth-year medical students. Osteoporosis, a critical and growing health concern, merits enhanced focus within undergraduate medical education, and our course was designed to address this need by consolidating knowledge distributed across several mandatory subjects.

We adopted a blended approach, integrating e-learning with case-based seminars that employ PBL methods, for several reasons: (1) contemporary medical students at our faculty prefer e-learning materials, (2) Problem and Case-Based Learning aligns well with modern educational theories emphasizing contextual and applied knowledge [13–22], (3) the authors of this article have had positive experiences with e-learning studies used by junior doctors preparing for specialized exams (Educational Center for Anatomy and Endoscopy: http://ecae.lf3.cuni.cz/ and the Center for Integrated Study

**Table 2. Elective Osteoporosis Course II test: Comparison of points acquired by Study Group II and Control Group II.**

| Number of points acquired | Number of students Study Group II ($N^{II}$) | Number of students Control Group II ($n^{II}$) |
|---|---|---|
| 11 | | 2 |
| 12 | | 2 |
| 13 | | 2 |
| 14 | | 1 |
| 15 | 1 | 2 |
| 16 | 2 | 1 |
| 17 | | 2 |
| 18 | 1 | 2 |
| 19 | 2 | |
| 20 | | 1 |
| 21 | 5 | |
| 22 | 7 | 2 |
| 23 | 2 | 3 |
| 24 | 4 | |
| 25 | | 1 |
| 26 | 1 | 2 |
| 27 | | 1 |
| 28 | 2 | |
| Total number of students | 27 | 24 |

Applying Bayes' approach, the resulting probability of success on the test is 99%. This means that students who completed the Elective Osteoporosis Course II would pass the test with a 99% probability.

of the Pelvis: http://medical-cisp.lf3.cuni.cz/). It turned out that osteoporosis was a suitable choice for creating an elective course, as more than 250 students have completed the course over its 12 years of existence.

The considerable difference in the test scores between Study Group I and Control Group I can be ascribed to the fact that the Elective Osteoporosis Course I was more theoretically oriented. Fifth-year medical students, who did not participate in the elective course (Control Group I) demonstrated less detailed knowledge of osteoporosis pathophysiology, diagnostics, and pharmacological treatment than students who engaged with the elective's structured, PBL-oriented content, indicating that participation in the elective contributed to improved knowledge retention.

Conversely, the smaller but still significant difference observed in the Elective Osteoporosis Course II (Study Group II vs. Control Group II) may be attributed to the clinical emphasis on this study phase. The sixth-year students who did not participate in the elective course (Control Group II) had already completed courses in Orthopedics and Traumatology of the musculoskeletal system, including case presentations of osteoporotic fractures, partially bridging the knowledge gap.

We acknowledge the potential influence of self-selection bias, as students opting into the elective course may have been more motivated or had prior interest in osteoporosis, which could have contributed to the observed differences. Additionally, the extra study time inherent to the elective course may have also contributed to the improved outcomes. This limitation should be taken into account when interpreting our findings.

Our results align with previous research supporting the efficacy of PBL in medical education [17–22]. Notably, our study highlights the value of elective, focused courses that complement core curricula by integrating knowledge across disciplines through active, case-based learning. Consistent with other authors [13–16], we emphasize the critical role of case study analysis in reinforcing clinical reasoning and knowledge retention. The superior performance of Control Group II relative to Control Group I likely reflects their increased exposure to clinical case-based learning in later years.

Although comparison with a traditional course was not possible in this study, we suggest that future research could involve a controlled comparison between PBL and traditional teaching approaches to better understand the relative efficacy of each method. Further studies might consider randomized controlled designs comparing PBL with traditional didactic teaching to better delineate relative benefits.

## Conclusion

This study demonstrates a higher degree of knowledge in medical students who, in addition to finishing the standard osteoporosis curriculum, completed an elective course—combining e-learning with problem-based interactive seminars. The e-learning component established essential theoretical groundwork, while the PBL format fostered active learning, critical thinking, and deeper understanding.

Our findings support PBL as an effective method for integrating comprehensive, up-to-date disease knowledge across diverse medical disciplines, which is particularly important for multifaceted conditions like osteoporosis. Elective courses can be a suitable complement to the traditional curriculum, especially for multidisciplinary medical topics.

Finally, these results reinforce the widely recognized advantage of incorporating case studies into both preclinical and clinical education to enhance long-term retention and application of theoretical knowledge.

## Supporting information

**S1 Graph. Elective Osteoporosis Course I test: Comparison of points acquired by control group I and study group I.**
(DOCX)

**S2 Graph. Elective Osteoporosis Course II test: Comparison of points acquired by control group II and study group II .**
(DOCX)



**S1 Data. Data for Graph 1 and 2.**
(DOCX)

**S1 Raw Data. Raw data Study I vs Control I.**
(XLSX)

**S2 Raw Data. Raw data Study II vs Control I.**
(XLSX)

## Author contributions

**Conceptualization:** Iva Hoffmanová, Valér Džupa, Václav Báča.

**Data curation:** Petr Waldauf.

**Formal analysis:** Iva Hoffmanová, Petr Waldauf, Robert Grill.

**Methodology:** Petr Waldauf, Václav Báča.

**Supervision:** Robert Grill.

**Validation:** Petr Waldauf.

**Visualization:** Petr Waldauf.

**Writing – original draft:** Valér Džupa.

**Writing – review & editing:** Iva Hoffmanová, Robert Grill, Václav Báča.

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
