## [Decision Letter · Decision Letter 0]

23 Jul 2025

Dear Dr. Hoffmanová,

Thank you for submitting your manuscript to PLOS ONE. After careful consideration, we feel that it has merit but does not fully meet PLOS ONE’s publication criteria as it currently stands. Therefore, we invite you to submit a revised version of the manuscript that addresses the points raised during the review process.

We look forward to receiving your revised manuscript.

Kind regards,

Muhammad Abbas Abid, MD

Academic Editor

PLOS ONE

Journal Requirements:

NO authors have competing interests

4. Please note that your Data Availability Statement is currently missing the DOI/accession number of each dataset OR a direct link to access each database. If your manuscript is accepted for publication, you will be asked to provide these details on a very short timeline. We therefore suggest that you provide this information now, though we will not hold up the peer review process if you are unable.

5. Please include your tables as part of your main manuscript and remove the individual files. Please note that supplementary tables (should remain/ be uploaded) as separate "supporting information" files

Additional Editor Comments:

The topic of the study is important as Osteoporosis is a multi-specialty condition often neglected. However, the manuscript and the methodology requires much greater rigor and explanation.

First, the authors state that osteoporosis is an emerging health problem but fail to mention how that is an emerging problem. A little detail about its prevalence and distribution will add more interest and relevance to the Introduction.

Similarly, the 2 groups need to be defined in much greater detail as there is overlap and discreet differences in the 2 groups. As the course was elective, and rightly so, it adds selection bias. Please mention why the course was elective (probably because it was not possible to make it mandatory), and mention it clearly in the limitations.

Please take a closer look at the reviewers' comments and address them all (specially the details shared by Reviewers 2, 4 and 5). Addressing this will not only add clarity to the study, but will also make it an interesting read.

I understand that there is quite some work required to turn it into a really interesting read but I believe it is worth a shot as the topic is both interesting and relevant.

Good luck to the authors and I applaud their efforts!

Reviewers' comments:

Reviewer's Responses to Questions

**Comments to the Author**

1. Is the manuscript technically sound, and do the data support the conclusions?

Reviewer #1: Yes

Reviewer #2: Partly

Reviewer #3: No

Reviewer #4: No

Reviewer #5: Partly

2. Has the statistical analysis been performed appropriately and rigorously?

Reviewer #1: Yes

Reviewer #2: Yes

Reviewer #3: I Don't Know

Reviewer #4: No

Reviewer #5: No

3. Have the authors made all data underlying the findings in their manuscript fully available?

Reviewer #1: Yes

Reviewer #2: Yes

Reviewer #3: Yes

Reviewer #4: Yes

Reviewer #5: Yes

4. Is the manuscript presented in an intelligible fashion and written in standard English?

Reviewer #1: Yes

Reviewer #2: Yes

Reviewer #3: Yes

Reviewer #4: Yes

Reviewer #5: Yes

Reviewer #1: Thank you for asking to review this manuscript.

It is a well conducted study related to Med Education and authors have indeed done a good job.

Aim is clearly mentioned and study is designed as per aim too.

They conduced the a very good analysis of both cohorts and described the results in a precise and legible way.

They have also discussed their outcomes very nicely with pertinent references.

Reviewer #2: The effect of Problem Based Learning (PBL) on medical students' learning outcomes is a very interesting area of research and the authors course design and outcomes are of interest and worthy of publication. However, the paper needs significant revision before it should be considered for publication.

More specific comments are included in the attached PDF, which has highlights and comments on the original paper. Particular care should be given to addressing the issues highlighted in red.

Here I will give general comments and advice for resubmission:

The authors have chosen to present this as a quantitative paper however they have not addressed several issues which arise due to the non-randomized selection of students into the study and control groups. This needs to be addressed.

Education is not a perfect science and even for a quantitative analysis of the outcomes more detail regarding the students motivation (for selection the course) and the background environment (what does it mean to "pass or fail" the course. etc.) need to be given in order for readers to be able to generalize the outcomes to their own educational situations.

Possibly, it would have been better to compare two elective courses, one taught with a PBL style and one with a more traditional approach.

It is suggested that the authors either address the issues above or reorient the paper to focus on a descriptive/qualitative view of the course before resubmitting the paper for publication.

Reviewer #3: My major concern with this study is that it is not a comparison of two instructional methods but rather it appears that both groups experienced the same [mandatory] curriculum while the study groups who used the elective had ADDITIONAL study of osteoporosis. Perhaps I misinterpreted how the authors presented the different study/control groups but it would seem logical that students with more and varied exposure to a problem would perform better on a knowledge test regarding that problem. I find it difficult to attribute better performance to the use of PBL under these circumstances. If the study groups did not participate in the mandatory osteoporosis curriculum then the comparison would be more legitimate.

Another difficulty I had in understanding the data presented was the fact that "test points" ranged from -x to +y. It is not clear to me how a test score could be negative [-].

Reviewer #4: The manuscript addresses an important topic in medical education by exploring the effects of an elective osteoporosis course using a problem-based learning (PBL) approach. The study is relevant given the interdisciplinary nature of osteoporosis and the increasing interest in active learning methods. While the research question is meaningful, several aspects of the study design and reporting require clarification or improvement, as detailed below.

1. The authors emphasize the use of PBL in the Elective Osteoporosis Course, which is a valuable teaching method. PBL focuses on student-centered learning, encouraging collaborative problem-solving in groups, which helps students develop critical thinking and problem-solving skills. However, the manuscript provides limited detail on the specific execution of PBL, including group configurations, lesson planning, and implementation processes. Providing more information on these aspects would help readers better understand how PBL was applied in this course.

2. If the authors intended to compare the elective and mandatory courses, they should more explicitly contrast their educational approaches, course content, and outcomes. For example, the authors could compare how the integration of osteoporosis into mandatory courses contrasts with the more focused approach of the elective course. Additionally, they could clarify if and how the elective course offers advantages in terms of student engagement, knowledge retention, and clinical application compared to the mandatory courses. This comparison would provide a clearer understanding of the relative effectiveness of the elective course in enhancing students’ knowledge of osteoporosis.

3. The manuscript does not clearly state whether Study Group I and Study Group II also completed the related mandatory courses on osteoporosis. This is essential information, as it affects the interpretation of the study results. If the study groups completed both mandatory and elective courses, the findings support the additive value of the elective course. If they only took the elective course, the results would suggest it may serve as an effective substitute or standalone intervention. Clarifying this point would strengthen the study’s conclusions.

4. In this study, all students in the study groups were in their fourth year, while the control groups consisted of fifth- and sixth-year students. This difference in academic level introduces significant confounding variables, such as variation in clinical exposure, exam preparation status, and recency of learning. These factors could affect knowledge retention and test performance independently of the elective course. The current comparison design limits the ability to attribute differences in outcomes solely to the elective course. I recommend that the authors acknowledge and discuss this limitation in the manuscript.

5. Since the elective course was likely voluntary, the study may be subject to self-selection bias. Students who chose to enroll in the elective may be more motivated or academically stronger, which could independently influence their test performance. I recommend the authors acknowledge and discuss this potential source of bias.

6. It appears that the test questions used to assess both the study and control groups were randomly selected from a larger pool of 100 items, meaning that each student received a different set of questions. While this approach increases variety and minimizes recall bias, it raises concerns about the comparability of scores across groups, as the difficulty level of each test version may vary. I recommend that the authors clarify whether any standardization or item difficulty balancing was applied.

7. The study would benefit from a more detailed description of the research procedure. Key aspects such as the timing of the assessments, the conditions under which the tests were administered, and whether the procedures were standardized across groups are not clearly stated. Additionally, information on participant selection or group assignment is lacking. Clarifying these points would help assess the internal validity of the study and the reliability of the comparisons made.

8. The manuscript states that the study does not involve human subjects and therefore did not require ethics approval. However, it involves collecting and comparing the test results of identifiable student groups under different instructional conditions, which typically falls within the scope of educational research involving human participants. I recommend that the authors clarify whether institutional ethics review was conducted or waived, and whether informed consent was obtained from participants.

9. The statistical analysis uses linear regression with robust standard errors, which is appropriate. However, the manuscript lacks details regarding covariate control or model assumptions. Were any variables such as academic year or prior GPA considered as potential covariates? Clarifying this would strengthen the interpretation of the results.

Reviewer #5: The authors conducted a study to investigate the effect of an Elective Osteoporosis Course on 4th year medical students' Osteoporosis related knowledge acquisition. It is a good attempt to renovate PBL with this self-directed learning driven elective course. However, several issues need to be addressed to improve the manuscript:

1. Major Concern: Study design is a major concern. Based on the description,

1) Study group 1 students seem to complete the Elective Osteoporosis Course on top of finishing all mandatory courses, while control group students only complete the mandatory courses. If so, it would not be a surprise that study group 1 students could have higher scores.

2) Study group 2 students differ from study group 1 students, which disrupts the ability to exam the continuity of course and its longitudinal effect on knowledge acquisition, given course 1 and course 2 are two parts of the Elective Osteoporosis Course.

3) The sample size is not large, leading to the results may be easily distorted by confounders (e.g. student aptitude). The authors should specify how they control confounders and provide effect size under Results.

2. Other Issues:

1) Study goal: The authors are suggested to specify the study hypothesis and the outcome measures of "depth of acquired knowledge".

2) Methods:

A) Please add a "Participants and Setting" section to help readers understand the context of this study.

B) The authors are recommended to illustrate the data collection and analysis process with a flowchart.

C) Please provide validity evidence of the knowledge measurement exam/tool. It is necessary to support the results are reliable and valid.

3) Results and Discussion:

A) The authors are suggested to discuss potential implication, such as whether the traditional mandatory courses would be replaced by this elective course.

B) Please add study limitations.

**Do you want your identity to be public for this peer review?** For information about this choice, including consent withdrawal, please see our Privacy Policy

Reviewer #1: **Yes: ** Dr Maseeh uz Zaman

Reviewer #2: **Yes: ** Francesco Bolstad

Reviewer #3: No

Reviewer #4: No

Reviewer #5: No

---

## [Author Response · Author response to Decision Letter 1]

5 Oct 2025

Dear Dr. Muhammad Abbas Abid and Reviewers,

Thank you very much for your thoughtful and constructive feedback on our manuscript titled „An Osteoporosis course as a Separate Component of Problem-Based Learning“

We appreciate the time and effort invested in reviewing our work and value the insightful comments that helps us improve the quality and clarity of the manuscript. Below we provide detailed responses to each point raised, along with descriptions of the changes made in the revised manuscript. In the revised version of the manuscript, the changes are highlighted in yellow.

General response to Additional Editor Comments

In the Introduction, we have added a more comprehensive introduction outlining the epidemiology and significance of osteoporosis as an emerging health problem, including recent prevalence data and the burden of disease.

In the Methods, subsection Participants and groups, we have clarified the definition and characteristics of the Study and Control groups.

In the Introduction, we have explained, why the course was elective.

Potential for self-selection bias are mentioned in the Methods, specifically in the end of Participants and group subsection, and in the Disscusion.

All reviewer comments have been addressed in detail below.________________________________________

Responses to Reviewers

Reviewer #2

Comments in the attached PDF:

We have addressed all comments, as detailed below:

Elective Course & Biases: We clarified the meaning of the Elective Course in the ‚Introduction‘. Potential biases are mentioned in the ‚Methods‘ section, specifically in the end of ‚Participants and group‘ subsection, and in the ‚Disscusion‘ section.

• Abstract:

1. Used optional “Elective Osteoporosis Course” to emphasize the elective nature of the course.

2. Corrected highlighted words.

3. Added the test scale in the parenthesis.

• Introduction: Provided a more detailed rationale for osteoporosis as a health issue and for the creation of the “Elective Osteoporosis Course”.

• Methods:

1. Explained the sequence of self-study e-learning followed by in-class sessions.

2. Addressed possible selection bias under Study Groups; omitted the word successfully for Study Group I.

3. Clarified that all Control Group I students completed all 4th-year and 5th-year osteoporosis-related courses.

4. Detailed the multiple-choice test methodology and passing criteria.

• Results:

1. Moved the test description and scoring from the ‚Results‘ to the ‚Methods‘ section; clarified credit conditions.

2. Explained clearly which students passed/failed and received/did not receive credit.

3. Added missing words.

• Conclusion: removed “unequivocally proved” and clarified, why Problem-Based Learning likely contributed to better knowledge retention.

General comments - major points:

1. Non-randomized selection and potential bias:

We acknowledge that the elective nature of the course introduces a self-selection bias as students with a particular interest or motivation in osteoporosis may be more likely to enroll. This is now explicitly stated in the ‚Methods‘ section, specifically in the end of ‚Participants and group‘ subsection, Unfortunately, randomization was not feasible given the educational context. We have revised the manuscript to emphasize this limitation and discuss its implications for generalizability in the ‚Disscusion‘ section.

2. Details on student motivation and background environment:

For greater clarity, we have newly divided the 'Methods' section into the following subsections: 'Course Structure', 'Assessment', and 'Participants and Groups'. We added following information in the 'Methods' section: The academic context is described in 'Course Structure' subsection. The criteria for passing or failing the course are described in 'Assessment' subsection. The factors influencing student motivation to take the elective course are mentioned in 'Participants and Groups' subsection.

3. Suggestion to compare two elective courses:

While comparing two elective courses with different teaching methods would be ideal, in this study only the osteoporosis elective was available as an additional course. In the ‚Discussion‘ section, we have noted this as a limitation and suggested it as an area for future research.

Reviewer #3

Major concerns:

1. Clarification on groups and additional exposure:

For greater clarity, we have newly divided the 'Methods' section into the following subsections: 'Course Structure', 'Assessment', and 'Participants and Groups'. We clarified in the 'Participants and groups' subsection, that all students completed the mandatory curriculum on osteoporosis, and the study groups additionally completed the elective course. This was added to the manuscript to highlight that the elective course was supplementary, which may explain the better test performance in study groups. We mentioned that as a limitation of the study in the 'Methods' section, specifically in the end of 'Participants and groups' subsection, and in the 'Discusion'.

2. Clarification on test scores and negative values:

In the 'Assessment' subsection, we described in detail how the multiple-choice final test was designed, including an explanation of why the achieved scores could range from negative to positive values

Reviewer #4

Detailed methodological clarifications requested:

1. Details on PBL implementation:

For greater clarity, we have newly divided the 'Methods' section into the following subsections: 'Course Structure', 'Assessment', and 'Participants and Groups'. In the 'Course Structure' we better described course design, with detailed information about the PBL implementation, facilitated by using case study learning. In the 'Participants and Groups' section are better described group sizes, and characteristics.

2. Comparison of elective vs mandatory courses:

In the section 'Participants and Groups' we now better clarified these differences and how the elective course complements the mandatory curriculum. Reasons for the better knowledge retention in PBL course are described in the subsection 'Course Structure', and also mentioned in the ‚Discusion‘.

3. Clarification of group composition: The manuscript now explicitly states that, all Study Group completed the mandatory courses in their study years. Additive value of the elective course is mentioned as a limitation study in the 'Methods' section, specifically in the end of 'Participants and groups' subsection, and in the 'Discusion'.

4. Academic level differences: are now better described in the 'Participants and groups' subsection, where the comparison of academic level is mentioned too. In the 'Discusion' is it also mentioned.

5. Self-selection bias acknowledgment:

We acknowledge that the elective nature of the course introduces a self-selection bias as students with a particular interest or motivation in osteoporosis may be more likely to enroll. This is now thoroughly discussed in the „Methods“ section, specifically in the end of „Participants and group“ subsection. We have revised the manuscript to emphasize this limitation in the „Disscusion“ section.

6. Test version comparability:

We clarified design of the final multiple-choice test in the 'Assessment' subsection, including the information, that each set of ten multiple-choice questions was designed to be equivalent in content and to maintain a numerically balanced distribution of response option.

7. Research procedure description:

We added detailed description of timing, test administration, and standardization of conditions in the ‚Assessment“ subsection.

8. Ethics and consent:

We obtained ethics committee approval for this educational research and ensured that all participants provided informed verbal consent. This is now clearly stated in the „Ethics Statement“ section.

9. Statistical analysis details:

In addition to the regression models, Bayes’ theorem was utilized to provide a more comprehensive statistical perspective

Reviewer #5

1. Major concerns:

For greater clarity, we have newly divided the 'Methods' section into the following subsections: 'Course Structure', 'Assessment', and 'Participants and Groups'. In the 'Participants and Groups' subsection, we clarified that the study groups completed both mandatory and elective courses, while controls completed only mandatory. We explained that all students in Osteoporosis Course II had previously completed Course I, while 2 students were newly included. We mentioned the limitations connected with this topic in the 'Discussion'.

2. Other Issues:

1. Study goal:

The hypothesis and primary outcomes (knowledge gain measured by test scores) are now explicitly stated in the 'Introduction' and 'Methods'.

2. Methods:

A) We added the 'Participants and Groups' subsection.

B) Instead of a flowchart, we provided a more detailed characterization of the study groups in the 'Participants and Groups' subsection..

C) Validity evidence of test tool:

In the 'Assessment' subsection we described the nature of the test and who developed and validated it.

3. Results and Discusion

A) Implication of the study are now better mentioned in the 'Conslusion'.

B) In the section 'Discussion', we decribed limitation of our study.

We hope that these revisions adequately address all concerns and improve the manuscript’s clarity, rigor, and overall quality. We thank the editor and reviewers again for their valuable feedback and look forward to your further consideration.

Sincerely,

Iva Hoffmanová, MD, PhD

Second Faculty of Medicine, Charles University

Prague, Czech Republic

---

## [Editor Report · Decision Letter 1]

2 Nov 2025

An Osteoporosis Course as a Separate Component of Problem-Based Learning

PONE-D-25-15083R1

Dear Dr. Hoffmanová,

We’re pleased to inform you that your manuscript has been judged scientifically suitable for publication and will be formally accepted for publication once it meets all outstanding technical requirements.

Kind regards,

Muhammad Abbas Abid, MD

Academic Editor

PLOS ONE
---

## [Editor Report · Acceptance letter]

PONE-D-25-15083R1

PLOS ONE

Dear Dr. Hoffmanová,

I'm pleased to inform you that your manuscript has been deemed suitable for publication in PLOS ONE. Congratulations! Your manuscript is now being handed over to our production team.

Kind regards,

on behalf of

Dr. Muhammad Abbas Abid

Academic Editor

PLOS ONE